# Management and clinical outcomes for patients with gastrointestinal bleeding who decline transfusion

Jessica O. Asiedu[1], Ananda J. Thomas[1], Nicolas C. Cruz[1], Ryan Nicholson[1], Linda M. S. Resar[2], Mouen Khashab[3], Steven M. Frank[1] *

1 Department of Anesthesiology and Critical Care Medicine, Johns Hopkins Medical Institutions, Baltimore, MD, United States of America, 2 Department of Hematology, Johns Hopkins Medical Institutions, Baltimore, MD, United States of America, 3 Department of Gastroenterology, Johns Hopkins Medical Institutions, Baltimore, MD, United States of America

* sfrank3@jhmi.edu

## Abstract

### Background

The national blood shortage and growing patient population who decline blood transfusions have created the need for bloodless medicine initiatives. This case series describes the management of gastrointestinal bleed patients who declined allogeneic blood transfusion. Understanding the effectiveness of bloodless techniques may improve treatment for future patients while avoiding the risks and cost associated with transfusion.

### Study design and methods

A retrospective chart review identified 30 inpatient encounters admitted between 2016 to 2022 for gastrointestinal hemorrhage who declined transfusion due to religious or personal reasons. Clinical characteristics and patient blood management methods utilized during hospitalization were analyzed. Hemoglobin concentrations and clinical outcomes are reported.

### Results

The most common therapy was intravenous iron (n = 25, 83.3%), followed by erythropoietin (n = 18, 60.0%). Endoscopy was the most common procedure performed (n = 23, 76.7%), and surgical intervention was less common (n = 4, 13.3%). Pre-procedure hemoglobin was <6 g/dL in 7 patients, and <5 g/dL in 4 patients. The median nadir hemoglobin was 5.6 (IQR 4.5, 7.0) g/dL, which increased post-treatment to 7.3 (IQR 6.2, 8.4) g/dL upon discharge. One patient (3.3%) with a nadir Hb of 3.7 g/dL died during hospitalization from sepsis. Nine other patients with nadir Hb <5 g/dL survived hospitalization.

### Conclusions

Gastrointestinal bleed patients can be successfully managed with optimal bloodless medi- cine techniques. Even patients with a nadir Hb <5–6 g/dL can be stabilized with aggressive

**Data Availability Statement:** Our anonymized data is available through BioStudies via accession number S-BSST1169 (https://nam02.safelinks. protection.outlook.com/?url=https%3A%2F%

2Fwww.ebi.ac.uk%2Fbiostudies%2Fstudies%2FS-BSST1169&data=05%7C01%7Cjasiedu3%40jhmi.edu%7Cb2fda42e41845911dfe08db99edd4cc%7C9fa4f438b1e6473b803f86f8aedf0dec%7C0%7C0%7C638273019012796990%7CUnknown%7CTWFpbGZsb3d8eyJWIjoiMC4wLjAwMDAiLCJQIjoiV2luMzIiLCJBTiI6Ik1haWwiLCJXVCI6Mn0%3D%7C3000%7C%7C%7C&sdata=UU9dEcdaxpWr9Zp8vhOanMpOidG9tu%2F3WVy2LXf0ebQ%3D&reserved=0).

**Funding:** Supported by a grant from the New York Community Trust (https://nam02.safelinks.protection.outlook.com/?url=https%3A%2F%2Fwww.nycommunitytrust.org%2F&data=05%7C01%7Csfrank3%40jhmi.edu%7Cb2144ba3dafe43749bca08db984b6f35%7C9fa4f438b1e6473b803f86f8aedf0dec%7C0%7C0%7C638271222446869035%7CUnknown%7CTWFpbGZsb3d8eyJWIjoiMC4wLjAwMDAiLCJQIjoiV2luMzIiLCJBTiI6Ik1haWwiLCJXVCI6Mn0%3D%7C3000%7C%7C%7C&sdata=03HPQVk9fuZSjvs2LejNrCp1Vr1AH4P4zncaB6CmomU%3D&reserved=0). Haemonetics provided support in the form of salaries for the author S.M.F. The company and their employees did not have any additional role in the study design, data collection and analysis, decision to publish, or preparation of the manuscript and only provided financial support in the form of authors' salaries and/or research materials. The specific role of this author is articulated in the 'author contributions' section.

**Competing interests:** S.M.F. serves on a scientific advisory board for Haemonetics, a company involved with patient blood management. This author does not participate in any patents, products in development, or marketed products from this company. This does not alter our adherence to PLOS ONE policies on sharing data and materials.

anemia treatment and may safely undergo anesthesia and endoscopy or surgery for diagnostic or therapeutic purposes. Methods used for treating bloodless medicine patients may be used to improve clinical care for all patients.

## Introduction

Allogeneic blood transfusion (ABT) is one of the most common hospital procedures in the U.S [1]. However, there is currently a growing population of patients who decline ABT due to religious or personal reasons. Members of the Jehovah's Witness religious group account for a major portion of this population [2]. In 1945, a doctrine was passed by Jehovah's Witnesses which prohibits its members from receiving ABT [3]. This is based on interpretation of certain passages in the Bible that equate a person's blood with their soul. An example of this is Leviticus 17:14 which states, "You must not eat the blood of any creature, because the life of every creature is its blood; anyone who eats it must be cut off" [4].

Additionally, while medical innovation has made ABT a relatively safe procedure, there are still many associated risks, such as transfusion-associated circulatory overload, transfusion-related acute lung injury, and acute hemolytic reactions [5]. These risks, along with the recent blood shortages that have resulted from the COVID-19 pandemic, and the continuing increase in healthcare costs associated with ABT have driven the need for alternative care known as patient blood management (PBM) [6]. Bloodless medicine is specialized care that combines a variety of PBM methods with the goal of avoiding ABT [3].

Bloodless medicine goes beyond simply withholding ABT, and includes various techniques utilized to reduce blood loss and stimulate red blood cell production. This includes methods such as the use of pediatric phlebotomy tubes to minimize blood loss from laboratory testing, and the treatment of anemia through agents like iron and erythropoietin [2]. These methods become even more critical in the management of patients suffering from active hemorrhage, especially with occult bleeding, as is often the case for patients suffering from gastrointestinal bleeding (GIB).

In this study, we retrospectively analyzed clinical management methods and clinical outcomes for GIB patients who declined transfusion. Moreover, methods used to efficiently manage these patients may be useful to reduce or avoid transfusions, even for patients who accept ABT.

## Materials and methods

The Johns Hopkins Medicine IRB granted approval for this study with waived informed consent under protocol # NA_00078426. We performed a retrospective chart review of patients who were admitted as inpatients to Johns Hopkins Hospital (JHH) between November 2016 and February 2022 for GIB and declined ABT. Eligible patients were identified from our bloodless center's database based on a primary diagnosis of GIB upon admission. Clinical data were subsequently retrieved from the patients' electronic charts via Epic (Verona, WI). Patients included in the study were all inpatients of at least 18 years of age who presented with overt signs of GIB (e.g., melena, hematochezia, and hematemesis). Patients who were seen multiple times for GIB were counted as separate encounters if admissions were at least three weeks apart. Three patients had two separate encounters during the reviewed timeframe such that our study encompasses 27 unique patients, and 30 patient encounters. Patients who were

initially admitted at external facilities for GIB, then transferred to JHH were excluded due to incomplete history of treatment and laboratory records prior to admittance at JHH.

Each patient was managed with a team of gastroenterologists, general surgeons, hematologists, and anesthesiologists who work together as a multidisciplinary team in the JHH Bloodless Medicine and Surgery program. Clinical decisions concerning surgical and medical interventions were made based on individual judgment of the medical team. The patient data examined included demographic factors like age, sex, BMI, baseline comorbidities, and anticoagulant usage. The elements of patient care and management that were analyzed were GIB type, therapeutic PBM interventions (e.g., iron, B12, folate, erythropoietin, and tranexamic acid), procedural or surgical interventions (e.g., endoscopy, colectomy), and clinical outcomes (e.g., length of stay, and mortality). Hemoglobin (Hb) concentrations are reported for the first, lowest (nadir) and last measurements before discharge.

Statistical analysis was performed via JMP v12 (SAS Institutes, Cary, NC). Continuous variables were described using measures of central tendency and outlier limits. Based on a non-normal distribution of hemoglobin concentrations, this variable was primarily analyzed using median and interquartile range (IQR). Categorical variables were measured as percentages. Comparison of continuous variables were performed using Wilcoxon signed-rank test.

## Results

From November 2016 to February 2022, 30 patients with GIB were identified that met the eligibility criteria for this study. Patient characteristics are summarized in Table 1. Of the 30 encounters, 10 (33.3%) were male patients and the average cohort age was 66 years. The most prevalent comorbidity in this population was hypertension, followed by diabetes mellitus. Other common comorbidities in this population were renal disease, congestive heart failure, cancer, and pulmonary disease. A substantial proportion of patients (70.0%) were using at least one anticoagulant at the time of their GIB, with a moderate percentage of patients (36.7%) on more than one anticoagulant. Aspirin was the most common anticoagulation therapy used in this cohort.

Diagnostic and therapeutic patient blood management techniques were utilized in these patient encounters to diagnose and treat anemia (i.e., iron deficiency anemia). Of the PBM therapeutics employed, IV iron (n = 25, 83.3%) and oral B12 and folate (n = 22, 73.3%) were the most common. Erythropoietin (n = 18, 60.0%), oral iron (n = 18, 60.0%), and IV B12 and folate (n = 18, 60.0%), were administered. Tranexamic acid was given to one patient (3.3%).

A Hb-based oxygen carrier (HBOC) Hemopure (HbO2 Therapeutics, Souderton, PA) was given to one patient (3.3%) on a compassionate use expanded access protocol. This patient had a nadir Hb of 3.5 g/dL (upon admission) and was showing signs of end-organ ischemia (ischemic bowel). This patient required a total colectomy and received 2 units of Hemopure prior to surgery when her Hb was 3.6 g/dL. She survived and was discharged with a Hb of 8.3 g/dL after a 31-day hospitalization, during which intravenous iron sucrose (200 mg daily for 11 days), intravenous erythropoietin (40,000 units twice daily for 12 days), and intramuscular B12 (1000 mcg/day), and intravenous folate (1 mg/day) were given.

In this cohort, 22 patients (76.7%) underwent endoscopy with the intention to locate and terminate the bleeding, for instance, through cauterization (Table 2). All endoscopies were performed with propofol anesthesia, and there was no peri-procedural adverse events or mortality. Four patients (13.3%) underwent colectomy or laparotomy to correct bleeding or excise cancerous tissue, all without perioperative adverse events or mortality. Two patients (6.7%) underwent CT angiography, and embolization of the inferior mesenteric artery was performed on one patient (3.3%). The pre-procedure hemoglobin was <6 g/dL in 7 patients, and <5 g/dL

**Table 1. Patient characteristics and therapeutic interventions for 30 gastrointestinal bleed patient encounters.**

|  | Parameter | N | Percent |
|---|---|---|---|
| **Patient Demographics** | Age, y | 66 ± 12* | —— |
|  | Sex (M) | 10 | 33.3 |
|  | BMI, kg/m$^2$ | 27.9 ± 8.2* | —— |
| **Comorbidities** | Hypertension | 20 | 66.7 |
|  | Diabetes Mellitus | 13 | 43.3 |
|  | Obesity | 3 | 30.0 |
|  | Congestive Heart Failure | 7 | 23.3 |
|  | Renal Disease | 9 | 30.0 |
|  | Pulmonary Disease | 6 | 20.0 |
|  | Cancer | 7 | 23.3 |
|  | Breast | 2 | 6.7 |
|  | Colon/Rectal | 1 | 3.3 |
|  | Liver | 2 | 6.7 |
|  | Pancreas | 1 | 3.3 |
|  | Prostate | 2 | 6.7 |
| **Anticoagulants** | Aspirin | 14 | 46.7 |
|  | Other NSAIDs | 4 | 13.3 |
|  | Warfarin/Coumadin | 4 | 13.3 |
|  | DOACs | 7 | 23.3 |
| **Patient Blood Management Methods** | Oral Iron | 18 | 60.0 |
|  | IV Iron | 25 | 83.3 |
|  | # of Iron Doses, median | 5 | —— |
|  | EPO | 18 | 60.0 |
|  | # of EPO Doses, median | 7.5 | —— |
|  | Oral B12 Folate | 22 | 73.3 |
|  | IV B12 Folate | 18 | 60.0 |
|  | Hemopure | 1 | 3.3 |
|  | Tranexamic Acid | 1 | 3.3 |

NSAID–Non-steroidal anti-inflammatory drugs, BMI–body mass index, DOAC–Direct-acting oral anticoagulant (includes apixaban, dabigatran, rivaroxaban, edoxaban), EPO–erythropoietin, IV–intravenous.

*Values represent mean ± SD

in 4 patients. Eleven cases (36.7%) were diagnosed as upper GIB, 14 (46.7%) were diagnosed as lower GIB, and in 5 cases (16.7%) the source of GIB could not be determined. The average length of stay was 11 days.

The median nadir Hb during hospitalization was 5.6 (IQR 4.5, 7.0) g/dL, which increased after treatment for anemia to 7.3 (IQR 6.2, 8.4) g/dL upon discharge (Fig 1). Ten patients had a nadir Hb <5.0 g/dL (Fig 2), and nine of them survived until discharge. Of the 30 cases, only one patient (3.3%) died during hospitalization. This patient had a nadir Hb of 3.7 g/dL, did not undergo anesthesia, endoscopy, or surgery, and death was attributed to septic shock in the setting of peritonitis. There were no additional deaths observed 30-days post-hospitalization.

## Discussion

In this study, we demonstrate that GIB patients can be successfully managed without the use of ABT when appropriate patient blood management methods are employed. This supports the results of previous cohort studies and case reports that demonstrate comparable mortality

**Table 2. Clinical characteristics and outcomes for 30 gastrointestinal bleed (GIB) encounters.**

|  | Parameter | N | Percent |
|---|---|---|---|
| **GIB Type** | Upper | 11 | 36.7 |
|  | Lower | 14 | 46.7 |
|  | Unspecified | 5 | 16.7 |
| **Procedure** | Endoscopy | 23 | 76.7 |
|  | Colectomy | 3 | 10.0 |
|  | Laparotomy | 1 | 3.3 |
|  | Embolization | 1 | 3.3 |
| **Pre-anesthesia Hb (g/dL)*** | <7 | 13/26 | 50.0% |
|  | <6 | 7/26 | 26.9% |
|  | <5 | 4/26 | 15.4% |
|  | <4 | 1/26 | 3.8% |
| **Clinical Outcomes** | Length of Stay, d (mean ± SD) | 11 ± 9 | —— |
|  | Mortality During Hospitalization | 1 | 3.3 |
|  | 30-day Mortality | 1 | 3.3 |

GIB–gastrointestinal bleeding. Hb–hemoglobin.

*Only 26 of the 30 patients underwent a procedure requiring anesthesia.

between patients with GIB that do and those that do not accept ABT [7, 8]. Although our sample size is small, the mortality rate of 3.3% represents an acceptably low rate compared to previous reports [7]. In cases with substantial blood loss such as with gastrointestinal hemorrhages, blood transfusions and blood products often serve as a life-saving procedure. Challenges arise however, when patients are unable to accept transfusions, as is the case for Jehovah's Witnesses. For these patients, the understanding and proper usage of patient blood management techniques are vital to their safe treatment and recovery.

In the United States, gastrointestinal hemorrhages are a relatively common occurrence with its reported annual incidence ranging from 50 to 100 persons per 100,000 [7]. GIB are usually classified as upper or lower GIB according to the origin of the bleed relative to the ligament of Treitz. An upper GIB refers to a bleed stemming anywhere from the esophagus to the duodenum while a lower GIB originates from the jejunum to the rectum [9]. Some risk factors associated with GIB are age and gender, with GIB more common in males and as one advances in age [9, 10]. Over the years, the mortality rate of GIB has remained relatively constant, averaging approximately 10%, despite advancements in medicine [7].

The bloodless management of GIB patients focuses on stopping the bleed, minimizing iatrogenic blood losses, and stimulating erythropoiesis. Blood conservation methods include the discontinuation of anticoagulants, and the substitution of adult phlebotomy tubes for pediatric tubes, which can result in ~70% reduction in blood loss from laboratory testing [11]. This is particularly important for hospitalized patients in the intensive care unit as they can experience a daily loss of up to 1% of their blood volume from phlebotomy alone [2]. Minimizing blood loss is arguably the most important facet of managing patients who do not accept ABTs in the setting of acute hemorrhage as it can eliminate the indication for transfusion.

Stimulation of erythropoiesis can be accomplished using recombinant human erythropoietin and supplements such as iron, vitamin B12, and folate [12]. Such agents generally produce optimal benefits when used long-term; however, their benefit in acute bleeds cannot be overlooked. In fact, a randomized control trial studying 74 patients with preoperative anemia undergoing valvular heart surgery demonstrated that the administration of a single

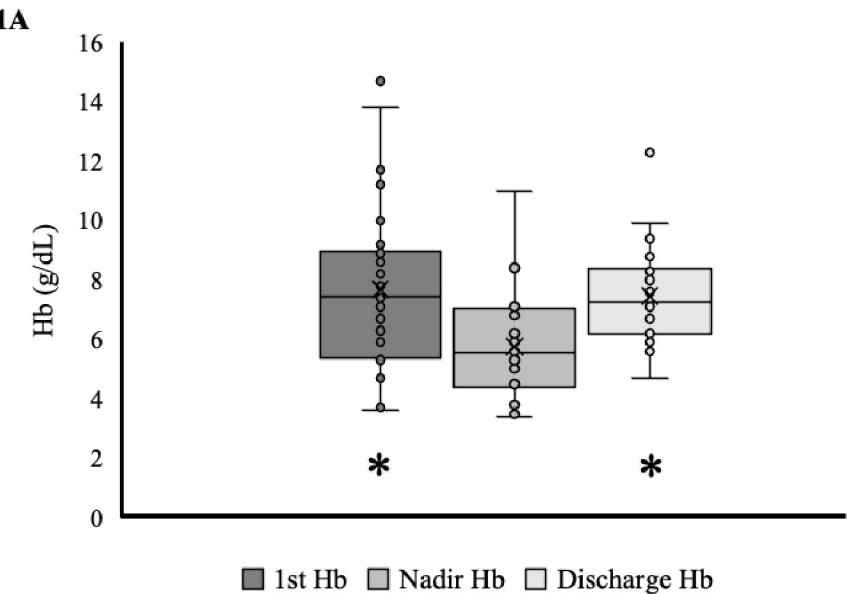

**1B**

|  | **1st Hb** | **Nadir Hb** | **Discharge Hb** |
|---|---|---|---|
| **Mean** | 7.7 | 5.8 | 7.5 |
| **Median** | 7.5 | 5.6 | 7.3 |
| **Min** | 3.6 | 3.4 | 4.7 |
| **Max** | 14.7 | 11.0 | 12.3 |

**Fig 1. First, nadir, and discharge hemoglobin (Hb) concentrations.** A) Boxplot of the first, nadir, and discharge Hb values of cohort. 'x' denotes the mean. B) Table identifying the mean, median, and outlier limits of the first, nadir, and discharge Hb concentrations. * denotes a statistically significant difference from the nadir median using Wilcoxon signed rank test.

intravenous dose of EPO and iron significantly reduces the need for perioperative transfusions [13]. In our study, various combinations of these therapies were utilized in patients to stimulate erythropoiesis, including the combination of iron and EPO. However, as medical insurance in some countries does not approve the use of EPO for the management of anemia from bleeding, it is important to consider if treatment with iron alone would be as effective. Some studies conclude that iron alone is sufficient to improve Hb levels in anemic patients with no additional benefit in the use of EPO, particularly in the setting of iron deficiency [14, 15]. Other work such as a metanalysis of 25 studies demonstrates that the combination of EPO and iron leads to a greater and faster increase in Hb in anemic patients compared to iron alone [16]. This lack of consensus in current literature demonstrates the need for further research in this area to better inform appropriate protocol in treating anemia in hemorrhagic patients.

Prior studies suggest that tranexamic acid (TXA) does not reduce mortality or bleeding events for GIB patients, however, it may increase the rate of venous thromboembolism and seizures [17, 18]. A recent large randomized control trial, HALT-IT, demonstrated that TXA usage in GIB does not reduce mortality and almost doubles the risk of venous thrombotic events [19]. As such, while TXA may moderately reduce mortality in patients at risk of

**2A**

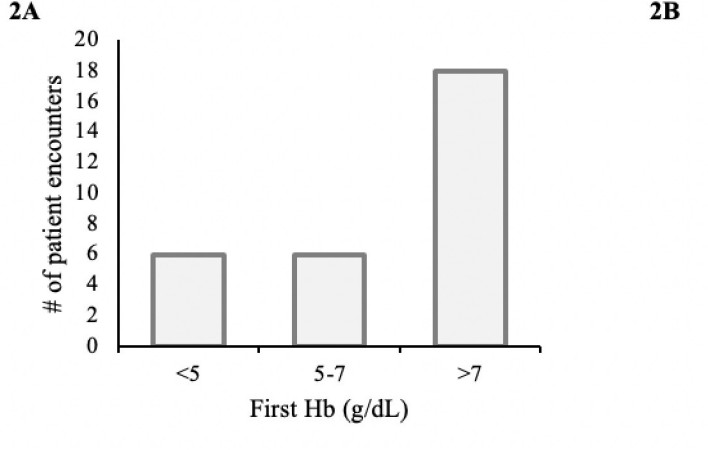

**2B**

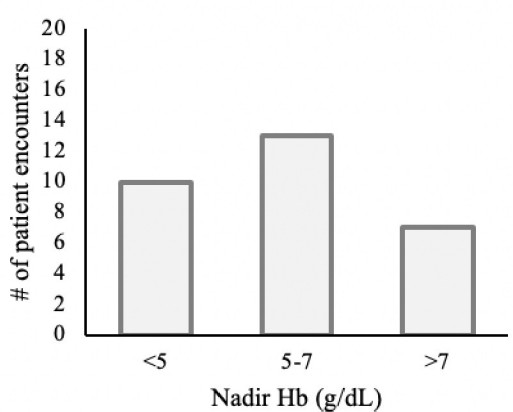

**2C**

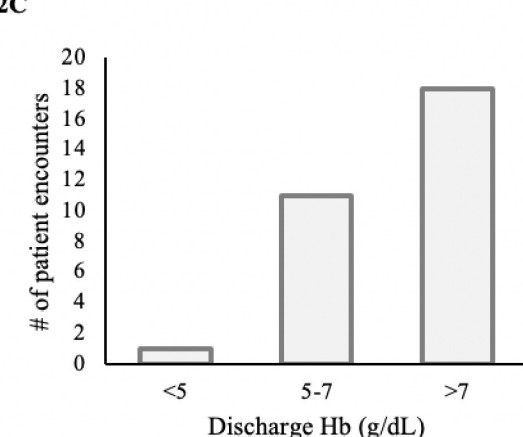

**Fig 2. Frequency of hemoglobin (Hb) concentrations during admission.** Number of patients within each range for Hb is reported upon admission (A), at the nadir Hb (B), and upon discharge (C).

hemorrhage secondary to trauma and postpartum hemorrhage [20, 21], it is not recommended for use in GIB patients [19]. In our case series, TXA was administered in only 1 of 30 cases. The decision to use TXA involves a risk/benefit decision analysis, and it is possible that for some patients who do not accept ABT who are severely anemic, the benefits of TXA may exceed the risks.

Although Jehovah's Witness patients will generally not accept red blood cells, plasma, platelets, or whole blood, other blood components and adjunctive therapies are considered acceptable, but by personal choice. One such therapeutic intervention is Hemopure, a bovine Hb-based oxygen carrier (HBOC), employed as an artificial oxygen carrier. The benefits of Hemopure are relatively immediate; delivery of one unit of Hemopure should increase a patient's Hb by 0.63g/dL [22]. As an experimental therapy, Hemopure is used on a compassionate use basis and was administered to one patient in our series. This patient had an admission Hb concentration of 3.6 g/dL and a nadir Hb of 3.5 g/dL. There were also signs of end-organ ischemia, including an elevated lactate level of 3.4 mmol/L and a low bicarbonate level of 17 mmol/L. After treatment, and a prolonged length of stay (31 days) with aggressive erythropoietic therapy, the patient was discharged with a Hb of 8.3 g/dL. Giving HBOCs also require a risk/benefit decision analysis, acknowledging that giving too little, too late may not help severely anemic

patients, while giving HBOCs for mild or moderate anemia may not be helpful to improve outcomes [1, 22, 23].

Another medical intervention that may be applicable to some Jehovah's Witness patients is autotransfusion, which may take place through autologous blood storage or salvage. Of note, the method of autotransfusion plays a key role in whether it complies with their religion as their practice denies the use of blood once it has left the body. As such, while preoperative autologous blood donation is generally not acceptable, these patients tend to accept the use of a cell saver or intraoperative autologous hemodilution during procedures with expected loss of a significant amount of blood. The latter procedures can be performed such that autologous blood remains physically contiguous with one's body [1, 2, 24]. Due to these intricacies, it is important to confirm the details of which autologous blood transfusions and blood products that each patient may accept.

Historically, a liberal transfusion strategy for GIB involved the administration of blood transfusion at a Hb concentration <10.0 g/dL; however, recent studies have shown that a more restrictive approach, using a Hb transfusion threshold <7.0 g/dL, results in a lower mortality for GIB cases [25]. As such, physicians are comfortable performing procedures with minimal expected blood loss (e.g., endoscopy) on patients with Hb >7.0 g/dL. However, previous studies have noted that mortality is substantially increased with a Hb <5.0 g/dL [26], which makes some anesthesiologists hesitant to provide anesthesia in such patients. For GIB patients, however, endoscopy serves as a critical diagnostic and treatment tool that can be used to localize and control hemorrhage [27]. Therefore, the ability and inclination to safely perform endoscopy is critically important in GIB patients with Hb concentrations <5.0 g/dL. In effect, this involves a risk/benefit decision, whereby the benefits of stopping the bleed typically outweigh the risks of anesthesia for endoscopy. In this study, 11 of 12 patients (91.7%) with a nadir Hb between 5.0–7.0 g/dL underwent endoscopy with no perioperative mortality. Additionally, 6 of 10 patients (60.0%) with a Hb <5.0 g/dL also underwent endoscopy without any peri-procedural mortality. This illustrates that with proper perioperative treatment of anemia, GIB patients can be stabilized and safely undergo procedures such as endoscopy to manage bleeding, even at Hb concentrations <5.0 g/dL.

## Limitations

The reliance on a primary diagnosis of GIB for selection of eligible patients excludes patients declining transfusion who may have developed GIB during their admission. This limits the sample size, reducing the statistical power of the study and its results. The use of this eligibility criteria, however, reduces confounding factors that might affect patient mortality and other clinical outcomes. While our relatively small sample size should be considered a limitation, the methods we report appear to be efficacious.

## Conclusion

With optimal bloodless medicine techniques, GIB patients can be successfully managed without ABT. With only one mortality in 30 cases, this represents an acceptable rate compared to previous reports [26]. Bloodless medicine emphasizes the respect of patients' rights and autonomy while providing effective care to improve patient outcomes. Previous studies have shown that bloodless medicine protocols result in similar or better clinical outcomes at equivalent or lower associated costs [3, 28, 29]. While bloodless medicine has historically been reserved for patients who decline transfusion for religious or personal reasons, the lessons learned from these patients can be applied to improve blood utilization and clinical care even for patients who accept ABT, thus avoiding unnecessary transfusions and promoting high-value care.

## Author Contributions

**Conceptualization:** Jessica O. Asiedu, Steven M. Frank.

**Data curation:** Jessica O. Asiedu, Ananda J. Thomas, Nicolas C. Cruz.

**Funding acquisition:** Steven M. Frank.

**Investigation:** Jessica O. Asiedu.

**Writing – original draft:** Jessica O. Asiedu, Steven M. Frank.

**Writing – review & editing:** Ananda J. Thomas, Nicolas C. Cruz, Ryan Nicholson, Linda M. S. Resar, Mouen Khashab.

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
