## [Decision Letter · Decision Letter 0]

16 Jun 2023

PONE-D-23-05326Management and Clinical Outcomes for Patients with Gastrointestinal Bleeding who Decline TransfusionPLOS ONE

Dear Dr. Frank,

Thank you for submitting your manuscript to PLOS ONE. After careful consideration, we feel that it has merit but does not fully meet PLOS ONE’s publication criteria as it currently stands. Therefore, we invite you to submit a revised version of the manuscript that addresses the points raised during the review process.

Please provide a point-by-point response to both reviewers' comments.

We look forward to receiving your revised manuscript.

Kind regards,

Mabel Aoun, MD, MPH

Academic Editor

PLOS ONE

Journal Requirements:

“S.M.F. serves on a scientific advisory board for Haemonetics.”

We note that one or more of the authors are employed by a commercial company: Haemonetics

3. Please include your full ethics statement in the ‘Methods’ section of your manuscript file. In your statement, please include the full name of the IRB or ethics committee who approved or waived your study, as well as whether or not you obtained informed written or verbal consent. If consent was waived for your study, please include this information in your statement as well

Reviewers' comments:

Reviewer's Responses to Questions

**Comments to the Author**

1. Is the manuscript technically sound, and do the data support the conclusions?

Reviewer #1: Yes

Reviewer #2: Yes

2. Has the statistical analysis been performed appropriately and rigorously? 

Reviewer #1: Yes

Reviewer #2: Yes

3. Have the authors made all data underlying the findings in their manuscript fully available?

Reviewer #1: Yes

Reviewer #2: Yes

4. Is the manuscript presented in an intelligible fashion and written in standard English?

Reviewer #1: Yes

Reviewer #2: Yes

5. Review Comments to the Author

Reviewer #1: Thank you for asking me to review this paper by Asiedu and colleagues. The authors have examined the management of patients admitted to their institution with GI bleeding who refused/declined red blood cell transfusions. Although most practitioners meet patients who would decline blood products, it remains relatively rare to face the situation of acute bleeding in this patient population. Although this series is small, it is nevertheless probably one of the largest and is informative. I have the following questions and comments for the authors:

-Please comment on the choice of therapies (IV iron, folate, B12), which are all important in stimulating erythropoiesis over relatively long period of time. These therapies cannot mitigate the short-term impact of acute blood loss and I am left wondering about strategies to mitigate blood loss that are unique to this patient population. Were other products were used? Was there any use of pro-coagulant factors or drugs, etc.

-Please clarify how many patients were seen multiple times and considered separate patients? Patients who are seen multiple times over a short period of time would be influenced/correlated to their previous admission (e.g. erythropoiesis would have already been stimulated, etc.).

-Please comment on the indications for surgery and endoscopy (as well as more generally for the whole cohort), and whether these interventions were necessary solely on the basis of blood products refusal. It is indeed very unusual to have to carry out colectomies or even therapeutic lower endoscopies to stop bleeding these days. Most cases of lower GI bleeding will resolve with supportive management. Unnecessary surgery or therapeutic endoscopies would indeed be a major endpoint to consider if invasive procedures could have been avoided with blood products. Bleeding from diverticulosis would be a good example of a pathology for which surgery is almost never indicated.

-Please include interventional radiology procedures in your description of invasive interventions.

-The results of this series should be put into context of the broader literature pertaining to acute blood loss management in patients who decline blood transfusions. There are surely other series that should be cited and reviewed. There may be systematic reviews as well?

Reviewer #2: This study focuses on the bloodless medicine approach to gastrointestinal bleeding patients who declined allogeneic blood transfusions. In addition, with the recent blood shortages resulting from the COVID-19 pandemic and the continuing increase in healthcare costs associated with allogeneic blood transfusions, the authors provide very important insight into reducing blood transfusions in all patients. Below are some points that need to be revised.

・Some types of blood transfusions may be acceptable for each patient

Jehovah’s Witness patients will generally not accept red blood cells, plasma, platelets, or whole blood, other blood components. In some patients, however, may accept autologous blood storage or closed extracorporeal circulation with autotransfusion using a cell saver during the procedure. These options should not be completely ruled out. It is important to confirm the details of precisely which autologous blood transfusions and blood products the patient can accept. This should be added to the DISCUSSION.

・Add to Reference

[Giving HBOCs also require a risk/benefit decision analysis, acknowledging that giving “too little, too late” may not help severely anemic patients, while giving HBOCs for mild or moderate anemia may not be helpful to improve outcomes.]

(DISCUSSION, page 12-13)

Please add references.

・Combination of intravenous iron and erythropoietin

In some countries, medical insurance does not cover the use of erythropoietin for anemia due to bleeding. In this study, some or many cases were treated with iron and erythropoietin. Is the increase in Hb greater with iron plus erythropoietin than with iron alone?

If possible, can the authors show the range of increase in Hb for each drug or combination used?

6. PLOS authors have the option to publish the peer review history of their article (what does this mean?). If published, this will include your full peer review and any attached files.

Reviewer #1: **Yes: **Guillaume Martel

Reviewer #2: No

---

## [Author Response · Author response to Decision Letter 0]

14 Jul 2023

We thank the Editor and Reviewers for the time and effort taken to suggest edits to improve our manuscript. Below is a point-by-point response addressing these revisions and describing how they were made. All edits in the revised manuscript are shown in underlined, red-colored font.

Reviewer #1

Comment- Please comment on the choice of therapies (IV iron, folate, B12), which are all important in stimulating erythropoiesis over relatively long period of time. These therapies cannot mitigate the short-term impact of acute blood loss and I am left wondering about strategies to mitigate blood loss that are unique to this patient population. Were other products were used? Was there any use of pro-coagulant factors or drugs, etc.

Response- We appreciate the reviewer’s insightful comment regarding the erythropoietic therapies we described in our manuscript and how these therapies do not acutely benefit the patient who is bleeding from the gastrointestinal tract. The same is true for erythropoietin as well. For this reason, the number one priority in patients who will not accept allogeneic transfusion is to stop the bleeding. We call this priority “keeping the blood in the patient”. After all possible efforts are made to stop the bleeding, then we focus on stimulating RBC production (erythropoiesis). Although the benefits of IV iron, folate, and B12 are not acute, we have seen increases in the hemoglobin concentration by 1-2 g/dL per week, which is important for achieving overall positive patient outcomes. Other products utilized in the management of these patients included tranexamic acid (an antifibrinolytic) and Hemopure (a bovine Hb-based oxygen carrier). These concepts are clarified and emphasized in the revised version of the manuscript in the edits on pages 12-14. 

Comment- Please clarify how many patients were seen multiple times and considered separate patients? Patients who are seen multiple times over a short period of time would be influenced/correlated to their previous admission (e.g. erythropoiesis would have already been stimulated, etc.).

Response- The reviewer raises a valid point about how previous therapeutic interventions for patients with recurrent GIB may influence their outcomes in subsequent encounters, particularly for encounters close in time when a patient may still be receiving some erythropoiesis-stimulating supplements. The number of patients who were seen multiple times for GIB during our study period have been noted on page 7 of the revised manuscript. Of note, two of the three repeat patients had encounters 9-13 months apart and are unlikely to be markedly influenced by previous therapeutic interventions.

Comment- Please comment on the indications for surgery and endoscopy (as well as more generally for the whole cohort), and whether these interventions were necessary solely on the basis of blood products refusal. It is indeed very unusual to have to carry out colectomies or even therapeutic lower endoscopies to stop bleeding these days. Most cases of lower GI bleeding will resolve with supportive management. Unnecessary surgery or therapeutic endoscopies would indeed be a major endpoint to consider if invasive procedures could have been avoided with blood products. Bleeding from diverticulosis would be a good example of a pathology for which surgery is almost never indicated.

Response- The reviewer raises an important topic which is the specific indication for surgery or endoscopy in our GIB patients. As we now emphasize in the revised version of the manuscript, and in the above response to reviewers’, the number one priority is stopping the bleed. If this can be done by endoscopy for example by cauterizing a bleeding source in the colon, this is a low risk, high benefit approach. Therefore, endoscopy is far more commonly utilized than surgery (e.g., colectomy). Another low risk, high benefit procedure is an interventional radiology procedure to embolize the source of bleeding. This is usually a second-choice approach after endoscopy. The third choice and as the reviewer alludes to, uncommonly performed, is surgical approach to stop the bleed. Rarely do we need to take these patients to surgery, in fact only four of our twenty-seven patients received such treatment. These details described here have been added to the revised version of the manuscript on page 10.

Comment- Please include interventional radiology procedures in your description of invasive interventions.

Response- Two out of twenty-seven patients in our case series required interventional radiology procedures. Although this approach to finding and stopping the bleed seems reasonable, it is not uncommon for the proceduralist to have difficulty finding the exact location and the anatomic vessels that need to be embolized to stop the bleed. Most likely stopping a GI bleed is more difficult that stopping bleeding from other sources such as the uterus in menorrhagia, since the vascular supply to the GI tract is so much more extensive and complex. Interventional radiology procedures have been described on page 10 and included in Table 2 of the revised manuscript.

Comment- The results of this series should be put into context of the broader literature pertaining to acute blood loss management in patients who decline blood transfusions. There are surely other series that should be cited and reviewed. There may be systematic reviews as well?

Response- In response to the above comment and on pages 12-14 of the manuscript, we have elaborated on some of the methods related to the management of blood loss in patient who decline blood transfusions. A lot of these methods revolve around minimizing blood loss and stimulating erythropoiesis. Some literature dive into other methods not discussed in our paper such as the use of hypotension to reduce blood loss or anesthesia to decrease oxygen consumption. While these methods are important to consider in the management of patients who decline transfusion, they are more applicable to patients undergoing major surgeries, which is not as reflective of our cohort. We have also included revisions to expand on how our results relate to previous studies done in this field.

Reviewer #2: 

Comment- Some types of blood transfusions may be acceptable for each patient

Jehovah’s Witness patients will generally not accept red blood cells, plasma, platelets, or whole blood, other blood components. In some patients, however, may accept autologous blood storage or closed extracorporeal circulation with autotransfusion using a cell saver during the procedure. These options should not be completely ruled out. It is important to confirm the details of precisely which autologous blood transfusions and blood products the patient can accept. This should be added to the DISCUSSION.

Response- The reviewer raises a valid point about the alternative blood products and components that may be acceptable to Jehovah’s Witness patients. In general, these patients do not accept primary blood components such as RBCs, platelets, or plasma. They may be more open to minor blood components like cryoprecipitate, albumin, clotting factors, but this determination is made on an individual basis. Some Jehovah Witness patients are also willing to accept autologous blood salvage and intraoperative hemodilution as these can be performed such that blood remains physically contiguous w/ one’s body. These details have been expounded upon in the revisions of page 14 of the edited manuscript.

Comment- Add to Reference.[Giving HBOCs also require a risk/benefit decision analysis, acknowledging that giving “too little, too late” may not help severely anemic patients, while giving HBOCs for mild or moderate anemia may not be helpful to improve outcomes.] (DISCUSSION, page 12-13). Please add references.

Response- As an experimental therapy, Hemopure is used on a compassionate use basis and requires a risk/benefit decision analysis of factors such as the severity of the patient’s anemia, timely and efficacious administration, and risk factors associated with administration such as increased vasoconstriction and hypertension. Such considerations have been further elaborated in the edits to the revised manuscript on page 14 with additional references (19, 20) cited for deeper exploration of this topic.

Comment- Combination of intravenous iron and erythropoietin

In some countries, medical insurance does not cover the use of erythropoietin for anemia due to bleeding. In this study, some or many cases were treated with iron and erythropoietin. Is the increase in Hb greater with iron plus erythropoietin than with iron alone?

If possible, can the authors show the range of increase in Hb for each drug or combination used?

Response- We appreciate the reviewer’s insightful comment about the use of EPO and iron for the correction of anemia and the role that medical insurance might play in the protocol to use either or a combination of both. Patients in our cohort received many different combinations of therapies. As such, it is difficult to isolate the contribution of any one therapy such as of iron, B12, or EPO. Moreover, current literature does not show a consensus concerning whether treatment of anemia is more efficacious with a combination of iron and EPO or iron alone. Some studies demonstrate that there is no difference in the requirement for transfusions when patients are treated with EPO and iron compared to iron alone. Other studies demonstrate that the addition of EPO to iron leads to a greater and faster increase in Hb in anemic patients compared to iron alone. As such, further studies are required to make any decisive conclusions about this, and we have not edited the manuscript based on this comment.

---

## [Decision Letter · Decision Letter 1]

31 Jul 2023

PONE-D-23-05326R1Management and Clinical Outcomes for Patients with Gastrointestinal Bleeding who Decline TransfusionPLOS ONE

Dear Dr. Frank,

Thank you for submitting your manuscript to PLOS ONE. After careful consideration, we feel that it has merit but does not fully meet PLOS ONE’s publication criteria as it currently stands. Therefore, we invite you to submit a revised version of the manuscript that addresses the points raised during the review process.

Please address the comment of Reviewer #2.

We look forward to receiving your revised manuscript.

Kind regards,

Mabel Aoun, MD, MPH

Academic Editor

PLOS ONE

Journal Requirements:

Reviewers' comments:

Reviewer's Responses to Questions

**Comments to the Author**

1. If the authors have adequately addressed your comments raised in a previous round of review and you feel that this manuscript is now acceptable for publication, you may indicate that here to bypass the “Comments to the Author” section, enter your conflict of interest statement in the “Confidential to Editor” section, and submit your "Accept" recommendation.

Reviewer #1: All comments have been addressed

Reviewer #2: All comments have been addressed

2. Is the manuscript technically sound, and do the data support the conclusions?

Reviewer #1: Yes

Reviewer #2: Yes

3. Has the statistical analysis been performed appropriately and rigorously? 

Reviewer #1: N/A

Reviewer #2: Yes

4. Have the authors made all data underlying the findings in their manuscript fully available?

Reviewer #1: Yes

Reviewer #2: Yes

5. Is the manuscript presented in an intelligible fashion and written in standard English?

Reviewer #1: Yes

Reviewer #2: Yes

6. Review Comments to the Author

Reviewer #1: (No Response)

Reviewer #2: [ ] are my new comments.

Reviewer #2:

Comment- Some types of blood transfusions may be acceptable for each patient

Jehovah’s Witness patients will generally not accept red blood cells, plasma, platelets, or whole blood, other blood components. In some patients, however, may accept autologous blood storage or closed extracorporeal circulation with autotransfusion using a cell saver during the procedure. These options should not be completely ruled out. It is important to confirm the details of precisely which autologous blood transfusions and blood products the patient can accept. This should be added to the DISCUSSION.

Response- The reviewer raises a valid point about the alternative blood products and components that may be acceptable to Jehovah’s Witness patients. In general, these patients do not accept primary blood components such as RBCs, platelets, or plasma. They may be more open to minor blood components like cryoprecipitate, albumin, clotting factors, but this determination is made on an individual basis. Some Jehovah Witness patients are also willing to accept autologous blood salvage and intraoperative hemodilution as these can be performed such that blood remains physically contiguous w/ one’s body. These details have been expounded upon in the revisions of page 14 of the edited manuscript.

[Thank you for correcting and adding references to alternative blood products and components. It is important to confirm these options for each Jehovah's Witness patient. Therefore, if possible, please add to the Discussion, "It is important to confirm the details of which autologous blood transfusions and blood products the patient can accept.]

Comment- Add to Reference. “Giving HBOCs also require a risk/benefit decision analysis, acknowledging that giving “too little, too late” may not help severely anemic patients, while giving HBOCs for mild or moderate anemia may not be helpful to improve outcomes.” (DISCUSSION, page 12-13). Please add references.

Response- As an experimental therapy, Hemopure is used on a compassionate use basis and requires a risk/benefit decision analysis of factors such as the severity of the patient’s anemia, timely and efficacious administration, and risk factors associated with administration such as increased vasoconstriction and hypertension. Such considerations have been further elaborated in the edits to the revised manuscript on page 14 with additional references (19, 20) cited for deeper exploration of this topic.

[Thank you for the correction. The references are added and will help the reader.]

Comment- Combination of intravenous iron and erythropoietin

In some countries, medical insurance does not cover the use of erythropoietin for anemia due to bleeding. In this study, some or many cases were treated with iron and erythropoietin. Is the increase in Hb greater with iron plus erythropoietin than with iron alone?

If possible, can the authors show the range of increase in Hb for each drug or combination used?

Response- We appreciate the reviewer’s insightful comment about the use of EPO and iron for the correction of anemia and the role that medical insurance might play in the protocol to use either or a combination of both. Patients in our cohort received many different combinations of therapies. As such, it is difficult to isolate the contribution of any one therapy such as of iron, B12, or EPO. Moreover, current literature does not show a consensus concerning whether treatment of anemia is more efficacious with a combination of iron and EPO or iron alone. Some studies demonstrate that there is no difference in the requirement for transfusions when patients are treated with EPO and iron compared to iron alone. Other studies demonstrate that the addition of EPO to iron leads to a greater and faster increase in Hb in anemic patients compared to iron alone. As such, further studies are required to make any decisive conclusions about this, and we have not edited the manuscript based on this comment.

[For the reader, it should be added in the Discussion with references that no conclusion has been reached as to whether the combination of EPO and iron or iron alone is more beneficial.]

7. PLOS authors have the option to publish the peer review history of their article (what does this mean?). If published, this will include your full peer review and any attached files.

Reviewer #1: **Yes: **Guillaume Martel

Reviewer #2: No

---

## [Author Response · Author response to Decision Letter 1]

2 Aug 2023

We thank the Editor and Reviewers for the time and effort taken to suggest edits to improve our manuscript. Below is a point-by-point response addressing these revisions and describing how they were made. All edits in the revised manuscript are shown in underlined, red-colored font.

Reviewer #1 (No additional comments)

Reviewer #2:

Comment- Some types of blood transfusions may be acceptable for each patient

Jehovah’s Witness patients will generally not accept red blood cells, plasma, platelets, or whole blood, other blood components. In some patients, however, may accept autologous blood storage or closed extracorporeal circulation with autotransfusion using a cell saver during the procedure. These options should not be completely ruled out. It is important to confirm the details of precisely which autologous blood transfusions and blood products the patient can accept. This should be added to the DISCUSSION.

Response- The reviewer raises a valid point about the alternative blood products and components that may be acceptable to Jehovah’s Witness patients. In general, these patients do not accept primary blood components such as RBCs, platelets, or plasma. They may be more open to minor blood components like cryoprecipitate, albumin, clotting factors, but this determination is made on an individual basis. Some Jehovah Witness patients are also willing to accept autologous blood salvage and intraoperative hemodilution as these can be performed such that blood remains physically contiguous w/ one’s body. These details have been expounded upon in the revisions of page 14 of the edited manuscript.

[Thank you for correcting and adding references to alternative blood products and components. It is important to confirm these options for each Jehovah's Witness patient. Therefore, if possible, please add to the Discussion, "It is important to confirm the details of which autologous blood transfusions and blood products the patient can accept.

Response- Thank you. This detail has been added to page 15]

Comment- Add to Reference. “Giving HBOCs also require a risk/benefit decision analysis, acknowledging that giving “too little, too late” may not help severely anemic patients, while giving HBOCs for mild or moderate anemia may not be helpful to improve outcomes.” (DISCUSSION, page 12-13). Please add references.

Response- As an experimental therapy, Hemopure is used on a compassionate use basis and requires a risk/benefit decision analysis of factors such as the severity of the patient’s anemia, timely and efficacious administration, and risk factors associated with administration such as increased vasoconstriction and hypertension. Such considerations have been further elaborated in the edits to the revised manuscript on page 14 with additional references (19, 20) cited for deeper exploration of this topic.

[Thank you for the correction. The references are added and will help the reader.]

Comment- Combination of intravenous iron and erythropoietin

In some countries, medical insurance does not cover the use of erythropoietin for anemia due to bleeding. In this study, some or many cases were treated with iron and erythropoietin. Is the increase in Hb greater with iron plus erythropoietin than with iron alone?

If possible, can the authors show the range of increase in Hb for each drug or combination used?

Response- We appreciate the reviewer’s insightful comment about the use of EPO and iron for the correction of anemia and the role that medical insurance might play in the protocol to use either or a combination of both. Patients in our cohort received many different combinations of therapies. As such, it is difficult to isolate the contribution of any one therapy such as of iron, B12, or EPO. Moreover, current literature does not show a consensus concerning whether treatment of anemia is more efficacious with a combination of iron and EPO or iron alone. Some studies demonstrate that there is no difference in the requirement for transfusions when patients are treated with EPO and iron compared to iron alone. Other studies demonstrate that the addition of EPO to iron leads to a greater and faster increase in Hb in anemic patients compared to iron alone. As such, further studies are required to make any decisive conclusions about this, and we have not edited the manuscript based on this comment.

[For the reader, it should be added in the Discussion with references that no conclusion has been reached as to whether the combination of EPO and iron or iron alone is more beneficial.

Response- This is indeed a crucial subject that may inform future protocol. These details have been expounded upon on page 13 with the appropriate references 14-16 added.]

---

## [Editor Report · Decision Letter 2]

7 Aug 2023

Management and Clinical Outcomes for Patients with Gastrointestinal Bleeding who Decline Transfusion

PONE-D-23-05326R2

Dear Dr. Frank,

We’re pleased to inform you that your manuscript has been judged scientifically suitable for publication and will be formally accepted for publication once it meets all outstanding technical requirements.

Kind regards,

Mabel Aoun, MD, MPH

Academic Editor

PLOS ONE
---

## [Editor Report · Acceptance letter]

15 Aug 2023

PONE-D-23-05326R2 

Management and Clinical Outcomes for Patients with Gastrointestinal Bleeding who Decline Transfusion 

Dear Dr. Frank:

I'm pleased to inform you that your manuscript has been deemed suitable for publication in PLOS ONE. Congratulations! Your manuscript is now with our production department. 

Kind regards, 

on behalf of

Dr. Mabel Aoun 

Academic Editor

PLOS ONE